# Variational Inference based Probabilistic Prompt for Rehearsal-Free Continual Learning

## Abstract

Continual learning aims to enable models to learn a sequence of tasks without catastrophic forgetting, a phenomenon where new information overwrites previously acquired knowledge. Traditional solutions for this problem, including regularization, replay buffers, and dynamic architectures, struggle with trade-offs in scalability, privacy, and adaptability. Prompt-based learning, initially developed in NLP, offers parameter-efficient alternatives by prepending learnable vectors to input tokens. However, existing prompt methods in continual learning, such as L2P and DualPrompt, rely on deterministic selection mechanisms that lack uncertainty modeling, making them less effective in dynamic and ambiguous task scenarios. In this work, we propose a novel framework called Variational Inference based Probabilistic Prompt (VPrompt) that introduces a stochastic latent variable formulation over prompt selection using variational inference. Our method learns an approximate posterior distribution over prompt assignments conditioned on inputs, and regularizes this with a uniform prior to ensure diversity and mitigate overconfidence. This probabilistic mechanism enables uncertainty-aware adaptation, improves robustness under domain shift, and eliminates the need for task labels or rehearsal buffers. We evaluate our method across Split CIFAR100, Split ImageNet-R, and a diverse 5-dataset benchmark. VPrompt consistently outperforms state-of-the-art baselines, including CODA-Prompt, L2P, DualPrompt, regularized and rehearsal-based methods, in terms of average accuracy and reduced forgetting. These results confirm that modeling uncertainty at the prompt level offers a scalable, buffer-free, and more flexible solution for continual learning.

## 1 Introduction

Continual learning (CL) Belouadah et al. (2021); De Lange et al. (2021); Masana et al. (2022); Van de Ven & Tolias (2019) refers to the ability of a model to learn from a sequence of tasks or data distributions without forgetting knowledge from earlier experiences. Unlike standard supervised learning, where the assumption is that all training data are available simultaneously and independently sampled, continual learning operates under sequential exposure to tasks, often with limited or no access to previous data. The primary research problem in continual learning is to achieve knowledge adaptation to new tasks while preserving prior knowledge, thus preventing what is known as catastrophic forgetting, the phenomenon where updating the model for a new task leads to abrupt performance degradation on previously learned tasks Lee et al. (2017); McCloskey & Cohen (1989); Mehta et al. (2023); Ramasesh et al. (2021).

Catastrophic forgetting remains a fundamental challenge for continual learning systems. When models are trained via standard gradient-based optimization, they tend to completely overwrite parameters important for previous tasks in favor of minimizing loss on the current task. As a result, maintaining a balance between stability (preserving old knowledge) and plasticity (adapting to new knowledge) becomes a critical requirement for successful continual learning.

To mitigate catastrophic forgetting, several major classes of solutions have been proposed: Regularization-based methods (e.g., Elastic Weight Consolidation Kirkpatrick et al. (2017), Synaptic Intelligence Zenke et al. (2017)) introduce penalties on parameter updates for important weights,

trying to preserve previously learned knowledge without relying on past data. Rehearsal-based methods (e.g., iCaRL Rebuffi et al. (2017), DER++ Buzzega et al. (2020)) store a buffer of samples from past tasks and replay them during training on new tasks. Dynamic architecture methods (e.g., Progressive Neural Networks Rusu et al. (2016)) expand the model's architecture to allocate task-specific modules.

While these approaches show varying levels of success, they face notable limitations. Regularization-based methods often struggle when task distributions are highly diverse, as simple penalties are insufficient for complex knowledge retention. Dynamic architectures become impractically large over long sequences of tasks. Rehearsal-based methods Bonicelli et al. (2022); Yoon et al. (2021), although highly effective, are increasingly recognized as problematic because they require storing raw data from previous tasks, raising privacy, scalability, and memory footprint concerns, especially in real-world or resource-constrained environments where data storage is not permissible (e.g., healthcare, robotics).

In this context, several core questions for advancing continual learning arise: First, how can models effectively adapt to new tasks without catastrophically forgetting old ones in the absence of stored data? Second, how can continual learners maintain flexibility without uncontrolled model expansion or excessive reliance on rigid regularization? Third, how can models explicitly account for uncertainty when deciding how to leverage past knowledge in new situations?

Our goal is to address these questions by developing a method that is rehearsal-free, parameter-efficient, and uncertainty-aware Gao et al. (2022); Liu et al. (2022b); Smith et al. (2021); Zhang et al. (2023); Liu et al. (2022a). Recently, prompt-based learning Smith et al. (2023); Tang et al. (2023); Pei et al. (2023) has emerged as a highly effective strategy in Natural Language Processing (NLP). Prompting techniques (e.g., prefix-tuning Li & Liang (2021), soft prompts Lester et al. (2021)) allow models to adapt to new tasks by prepending learnable input vectors while keeping the large pre-trained backbone models frozen. This reduces the number of trainable parameters and enables rapid adaptation. Inspired by this success, prompt-based methods like Learning to Prompt (L2P) Wang et al. (2022c) and DualPrompt Wang et al. (2022b) have been adapted to the vision domain for continual learning. However, applying prompt-based techniques to continual learning introduces unique challenges. Unlike NLP tasks where task identities and boundaries are often clear, in continual learning tasks in vision, the task information is ambiguous, overlapping, and often not explicitly provided. Static prompt selection strategies, as used in L2P and DualPrompt, do not model uncertainty and ambiguity during task transitions, making them vulnerable to forgetting in harder continual learning scenarios with domain shifts and ambiguous inputs.

To overcome these limitations, we propose a novel method called variational inference-based probabilistic prompting (VPrompt) for rehearsal-free continual learning. Rather than deterministically selecting or fusing prompts, our method models the selection of prompts as a latent probabilistic variable. Using a variational inference framework, we maintain an approximate posterior over prompts conditioned on the current input, and regularize it with a prior distribution to control prompt usage entropy. This enables the model to capture uncertainty in task identification and prompt relevance, allowing for better generalization across tasks while mitigating forgetting, without requiring any task labels or memory buffers.

Our work makes three key contributions:

- Variational Probabilistic Prompting: We introduce a novel variational inference-based probabilistic prompt selection mechanism for continual learning, explicitly modeling uncertainty in prompt usage to improve adaptability and memory retention without replay buffers.

- Unified Framework for Continual Learning: We integrate probabilistic prompting into a simple, scalable architecture based on pre-trained Vision Transformers (ViT), achieving rehearsal-free continual learning without fine-tuning the base model.

- Extensive Benchmarking: We conduct comprehensive experiments across Split CIFAR100, Split ImageNet-R, and a heterogeneous 5-dataset benchmark, demonstrating superior performance in terms of accuracy and reduced forgetting compared to prompt-based, replay and regularization-based state-of-the-art methods.

## 2 RELATED WORK

### 2.1 FINE-TUNING PRETRAINED MODELS

In recent years, the rise of pretrained *foundation models*, especially large-scale models like BERT Devlin et al. (2019), GPT Radford et al. (2019), and ViT Dosovitskiy et al. (2020), has revolutionized the field of machine learning. These models are initially trained on massive datasets to learn general-purpose representations, which can later be transferred to downstream tasks. The most straightforward method for leveraging such models is fine-tuning, where the entire pretrained network is updated using the labeled data of the target task Chung et al. (2024); Touvron et al. (2023); Liu et al. (2023a). While this approach often leads to strong performance, it has significant drawbacks: it is computationally intensive, requires storing and updating all model weights for each task, and tends to suffer from catastrophic forgetting when applied in a continual learning setting.

### 2.2 PROMPT TUNING

To address some of these limitations, the concept of prompt tuning Lester et al. (2021); Li & Liang (2021) has been introduced. Instead of updating the entire model, prompt tuning freezes the pretrained backbone and learns a small set of input embeddings, called prompts, that guide the model towards solving a specific task Liu et al. (2023b). These prompts can be thought of as learnable tokens prepended to the input, effectively modulating the model's internal attention mechanisms. This approach significantly reduces the number of trainable parameters and memory footprint, enabling more efficient adaptation to new tasks. Prompt tuning has shown remarkable success in natural language processing and, more recently, in vision-language and purely vision domains using architectures like ViT Jia et al. (2022); Khattak et al. (2023).

### 2.3 CONTINUAL LEARNING

Continual learning (CL) Wang et al. (2024); Ebrahimi et al. (2020), also known as lifelong learning Sarfraz et al. (2025); Zhao et al. (2022), is a setting where models learn from a sequence of tasks without forgetting previously acquired knowledge. This setting is fundamentally different from the standard supervised learning paradigm, which assumes access to the full dataset at once. In continual learning, data from previous tasks is no longer accessible after training. A major challenge in CL is catastrophic forgetting, where the model's performance on earlier tasks significantly degrades as it adapts to new ones Serra et al. (2018). This challenge is exacerbated in deep networks due to weight sharing across tasks. Various approaches to mitigate forgetting include regularization Kirkpatrick et al. (2017); Zenke et al. (2017), memory replay Rebuffi et al. (2017); Buzzega et al. (2020), and architectural methods such as dynamically expanding the model Rusu et al. (2016); Van de Ven et al. (2022).

**Regularization-based continual learning** techniques introduce constraints to preserve previously learned knowledge. Elastic Weight Consolidation (EWC) Kirkpatrick et al. (2017) estimates the importance of each parameter to previous tasks using the Fisher Information Matrix and penalizes significant changes during subsequent updates. Synaptic Intelligence (SI) Zenke et al. (2017) builds on this idea by accumulating task-relevant parameter importance over time using an online approximation. While effective, these methods often struggle when task boundaries are unclear or when tasks share overlapping distributions, limiting their flexibility in real-world settings. On the other hand, LwF Li & Hoiem (2017) proposes a framework that employs the knowledge distillation to regularize the network to mitigate the catastrophic forgetting problems.

**Rehearsal-based continual learning** methods address forgetting by maintaining a memory buffer of data from past tasks. iCaRL Rebuffi et al. (2017) and GDumb Prabhu et al. (2020) periodically replay this data during training to retain performance on previous tasks. Some approaches augment this buffer with synthetic data or prototypes, further enhancing generalization Buzzega et al. (2020); Cha et al. (2021). Although highly effective in practice, rehearsal-based methods often violate privacy or memory constraints, especially in scenarios where storing past data is infeasible or prohibited.

**Architecture-based continual learning** focuses on dynamically modifying the network structure. Progressive Neural Networks Rusu et al. (2016) expand the architecture by allocating new subnet-

works for each task, preventing interference at the cost of unbounded model growth. Other methods selectively activate task-specific submodules or freeze parts of the network Parisi et al. (2019); Wang et al. (2022a). While this approach eliminates forgetting through modularization, it poses scalability concerns and often requires explicit task labels.

## 2.4 Prompt Tuning for Continual Learning

Prompt tuning presents an appealing alternative in the continual learning landscape. By freezing the backbone and learning task-specific prompts, it allows for the isolation of task information while maintaining a shared representation space.

*Learning to Prompt (L2P)* Wang et al. (2022c) is a pioneering framework that merges prompt tuning with continual learning. L2P introduces a learnable prompt pool and employs an attention mechanism over learned keys to select relevant prompts for each input. This formulation enables task-conditioned inference using a frozen ViT backbone, reducing interference between tasks and achieving strong performance on standard CL benchmarks such as Split CIFAR100 and ImageNet-R.

However, L2P's prompt selection strategy is inherently deterministic, relying on nearest-neighbor key-query attention. This design choice ignores uncertainty in prompt-task relationships and limits the model's capacity to handle ambiguous or overlapping tasks. Our work addresses this limitation by introducing a probabilistic perspective to prompt selection, thereby enhancing both flexibility and robustness in non-stationary settings. Other work like progressive prompts Razdaibiedina et al. (2023) learns a new soft prompt for each task and sequentially concatenates it with the previously learned prompts, while keeping the base model frozen. DualPrompt Wang et al. (2022b) proposes to learn general prompt and task-specific prompts to insert into transformer layers to ensure the continual learning process. CODA prompt Smith et al. (2023) proposes to use attention mechanism to learn prompt components that can be used to contribute to the prompt learning process with a classification loss as the end to end learning framework. However, the stochastic processes of prompt selection has not been employed in these methods.

## 2.5 Towards Probabilistic Prompting

This work proposes extending prompt selection with a probabilistic framework grounded in variational inference. In this formulation, the selection of prompts is modeled as a stochastic process over a learned distribution, where prompt keys define a latent space over which uncertainty can be quantified. This formulation not only maintains modularity and parameter efficiency but also introduces uncertainty-awareness in prompt selection. This helps the model make more cautious updates and better handle task ambiguity, particularly useful in non-stationary environments.

## 3 Method

Continual learning (CL) requires models to learn from a sequence of tasks $\mathcal{D}_1, \mathcal{D}_2, \ldots, \mathcal{D}_T$ without catastrophic forgetting, the degradation in performance on previously learned tasks. In traditional setups, neural networks tend to overwrite prior knowledge when fine-tuned on new data, especially under task-agnostic conditions where no explicit task identifiers are given during inference.

The Learning to Prompt (L2P) framework Wang et al. (2022c) addresses this by freezing the backbone (e.g., Vision Transformer, ViT) and attaching a pool of learnable prompts, denoted as $\mathcal{P} = \{ P_1, \ldots, P_M \}$. Here $M$ represents the predefined number of prompts. For a given input $x$, prompts are selected based on nearest neighbors in a learned key space, and the selected prompts are prepended to the input tokens before being passed through the transformer.

In traditional Learning to Prompt (L2P) methods for continual learning, prompt selection is performed using a deterministic mechanism: for each input, the top-$k$ most similar prompt keys (often based on dot-product similarity with the input representation) are selected from a fixed pool and used to guide the model's representation. This method has proven effective in enabling parameter-efficient adaptation across tasks. However, its deterministic nature can lead to several limitations. As tasks change over time, fixed prompt selection strategies may overfit to early-task distributions or fail to adapt to subtle shifts in the input space. This inflexibility exacerbates catastrophic forgetting, where knowledge acquired from earlier tasks degrades as new ones are learned.

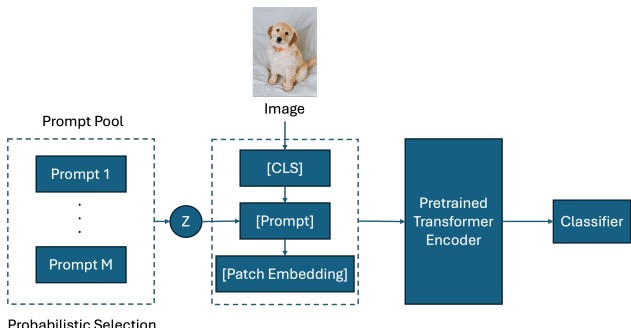

Figure 1: Our Variational-based Probabilistic Prompt framework for the rehearsal-free continual learning. An input (e.g., image) is passed through a query function (we use the pretrained ViT encoder) and used for a novel condition to formulate a probabilistic selection procedure for the prompts in the prompt pool. The weighted prompts are aggregated into a single prompt to prepend onto the input embeddings as the input to the classifier (the task head). Our prompt method is parameter efficient and no training data is stored for replay which is memory efficient and privacy preserving. Importantly, our probabilistic prompt selection scheme can be optimized end-to-end.

Table 1: Comparison between L2P and our probabilistic prompting method

| Feature | Traditional L2P | Probabilistic Prompting (Ours) |
|---|---|---|
| Prompt Selection | Deterministic (Top-K NN) | Probabilistic (Latent variable) |
| Uncertainty | Not modeled | Explicitly modeled via entropy of $q(z\|x)$ |
| Prompt Usage | Fixed | Input-conditional and adaptive |
| Regularization | Hard boundary per task | Smooth distribution with KL regularization |
| Knowledge Sharing | Limited across tasks | Encouraged through latent space reuse |
| Continual Learning | Effective but rigid | Flexible and uncertainty-aware |

To address these limitations, we propose a variational formulation of prompt selection (See Figure 1), where the choice of prompts is treated as a stochastic process governed by a learned probability distribution. Specifically, instead of selecting prompts based solely on fixed similarity scores, we define a latent variable $z$ that determines which prompts are used for a given input. We then learn an approximate posterior distribution $q(z|x)$, which reflects the model's belief over relevant prompts given input $x$. This posterior is typically parameterized using a Softmax function over the similarity scores between input representations and prompt keys. A prior distribution $p(z)$, often chosen as a uniform distribution over the prompt pool, is also defined to encourage even prompt utilization and avoid mode collapse. Specifically, we introduce a discrete latent variable $z \in \{1, \ldots, M\}$, representing the index of the prompt used for input $x$. Instead of hard-selecting prompts, we learn a distribution over prompts $q(z|x)$ and define the model as:

$$p(y|x) = \sum_{z=1}^{M} p(y|x, P_z) \cdot q(z|x).$$

Here, $y$ represents the labels that we are predicting. This probabilistic approach enables soft prompt routing and introduces variational uncertainty into the prompt selection mechanism, aligning with the demands of continual learning.

### 3.1 VARIATIONAL FORMULATION

Since the marginal likelihood $\log p(y|x)$ involves an intractable sum over prompts, we use variational inference (VI) to approximate it. We define the evidence lower bound (ELBO):

$$\log p(y|x) \geq \mathbb{E}_{q(z|x)}[\log p(y|x, z)] - \mathrm{KL}(q(z|x)\|p(z)),$$

where $q(z|x)$ is the variational posterior, modeled as a softmax distribution over learned similarity scores between the input embedding and prompt keys, $p(z)$ is a prior over prompts, typically uniform

to encourage general usage, and KL is the Kullback-Leibler divergence, acting as a regularizer. The first term encourages good performance by weighting prompt-conditioned predictions. The second term regularizes $q$ toward uniformity to prevent overfitting and collapsing to few prompts. This allows the model to make uncertainty-aware decisions about which prompts to use, avoid hard-coded task boundaries, and learn shared structure in prompt usage across tasks, crucial for continual learning. We compute $q(z|x)$ using:

$$q(z|x) = \text{Softmax}\left(\frac{f(x)^T K}{\tau}\right),$$

where $f(x)$ is the normalized embedding of input $x$, $K \in \mathbb{R}^{M \times d}$ is the prompt key matrix (also normalized), and $\tau$ is a temperature parameter. This soft distribution allows the model to assign varying levels of confidence to each prompt, encouraging shared use of prompts across tasks and more robust generalization. This probabilistic extension introduces multiple benefits for continual learning: uncertainty-aware prompt selection (inputs from ambiguous or transitional regions between tasks can use a mixture of prompts, smoothing the decision boundary and reducing abrupt forgetting), adaptive knowledge sharing (prompts are reused across tasks through soft allocation, helping to retain knowledge without task-specific overfitting), mitigation of overconfidence (by regularizing $q(z|x)$ via KL divergence, the model avoids collapsing to a few prompts, a common issue in deterministic L2P), and dynamic generalization (prompt distributions evolve as tasks arrive, enabling the model to discover emerging similarities between tasks and adaptively merge knowledge). Table 1 contrasts L2P and our probabilistic prompting method.

## 4 EXPERIMENTAL SETUP

### 4.1 DATASETS

To evaluate the effectiveness of our variational probabilistic prompting framework for continual learning, we perform experiments on three benchmark settings that span class-incremental, domain-incremental, and cross-domain generalization challenges. The three benchmarks are: (1) Split CIFAR100, (2) Split ImageNet-R, and (3) a heterogeneous 5-dataset sequence composed of SVHN, MNIST, CIFAR10, NotMNIST, and FashionMNIST. Each of these datasets introduces a unique domain with non-overlapping label spaces and significant variations in data distributions, rendering the benchmark highly challenging. We treat each dataset as a separate task in a 5-task continual learning sequence. All images are resized to 224×224 and converted to RGB if needed. This benchmark is particularly well-suited for assessing the effectiveness of probabilistic prompts in capturing task-specific nuances across disjoint visual modalities. These datasets were selected for their diversity in semantics, visual styles, and domain shifts, providing a comprehensive testbed for assessing catastrophic forgetting, prompt generalization, and task adaptation under uncertainty.

### 4.2 BASELINES

To rigorously evaluate the performance of our probabilistic prompting framework, we compare it against a range of state-of-the-art continual learning baselines spanning multiple methodological categories. These baselines include prompt-based, regularization-based, and rehearsal-based approaches, each offering different strengths and assumptions regarding continual learning.

The main baselines that we consider include: CODA-Prompt Smith et al. (2023), Learning to Prompt (L2P) Wang et al. (2022c), DualPrompt Wang et al. (2022b), Elastic Weight Consolidation (EWC) Kirkpatrick et al. (2017), Learning without Forgetting (LwF) Li & Hoiem (2017), Synaptic Intelligence (SI) Zenke et al. (2017), iCaRL (Incremental Classifier and Representation Learning) Rebuffi et al. (2017), DER++ (Dark Experience Replay) Buzzega et al. (2020), Co2L (Contrastive Continual Learning) Cha et al. (2021).

By benchmarking against these baselines, we ensure a comprehensive evaluation across variations in continual learning paradigm, prompt selection, weight regularization, knowledge retention, and replay. Our goal is to demonstrate that modeling uncertainty in prompt selection via variational inference offers consistent improvements, especially in scenarios involving high domain shift or ambiguous class boundaries.

Table 2: Results on Split-Imagenet-R dataset. Results are included for 5 tasks (40 classes per task), 10 tasks (20 classes per task), and 20 tasks (10 classes per task). $A_N$ represents the average accuracy across tasks and $F_N$ represents the average forgetting. We report results over 3 trials.

| Tasks | 5 | | 10 | | 20 | |
|---|---|---|---|---|---|---|
| Metrics | $A_N(\uparrow)$ | $F_N(\downarrow)$ | $A_N(\uparrow)$ | $F_N(\downarrow)$ | $A_N(\uparrow)$ | $F_N(\downarrow)$ |
| UB | 79.13 | - | 79.13 | - | 79.13 | - |
| iCaRL | $65.38 \pm .71$ | $22.28 \pm .67$ | $62.30 \pm .55$ | $25.54 \pm .88$ | $59.55 \pm .85$ | $22.74 \pm .33$ |
| DER++ | $69.11 \pm .45$ | $18.87 \pm .35$ | $66.73 \pm .87$ | $20.67 \pm 1.24$ | $64.45 \pm .34$ | $23.35 \pm .55$ |
| Co2L | $67.38 \pm .25$ | $20.28 \pm .62$ | $65.90 \pm .14$ | $23.36 \pm .71$ | $61.12 \pm .93$ | $28.86 \pm .26$ |
| SI | $40.55 \pm .79$ | $51.15 \pm .29$ | $37.76 \pm .95$ | $54.43 \pm .55$ | $35.52 \pm .69$ | $57.73 \pm .31$ |
| EWC | $38.33 \pm .56$ | $54.43 \pm .98$ | $35.00 \pm .43$ | $56.16 \pm .88$ | $31.67 \pm .45$ | $59.95 \pm .85$ |
| LwF | $40.86 \pm .43$ | $50.22 \pm .37$ | $38.54 \pm 1.23$ | $52.37 \pm .64$ | $31.44 \pm 1.15$ | $57.25 \pm .78$ |
| L2P | $63.25 \pm .55$ | $4.48 \pm .33$ | $61.14 \pm .35$ | $5.35 \pm .13$ | $59.33 \pm .43$ | $9.73 \pm .25$ |
| DualPrompt | $70.22 \pm .27$ | $\mathbf{4.13 \pm .23}$ | $68.13 \pm .49$ | $\mathbf{4.68 \pm .20}$ | $64.55 \pm .44$ | $\mathbf{5.92 \pm .19}$ |
| CODA-Prompt | $71.51 \pm .38$ | $4.99 \pm .19$ | $70.45 \pm .56$ | $7.64 \pm .10$ | $66.37 \pm 1.19$ | $9.96 \pm .15$ |
| VPrompt(Ours) | $\mathbf{75.80 \pm .45}$ | $4.41 \pm .17$ | $\mathbf{72.04 \pm .35}$ | $7.81 \pm .20$ | $\mathbf{69.70 \pm .31}$ | $9.81 \pm .16$ |

Table 3: Results on Split-Cifar100 and 5-datasets dataset. $A_N$ represents the average accuracy across tasks and $F_N$ represents the average forgetting. We report results over 3 trials. The Cifar100 results are included for 10 tasks (10 classes per task).

| Datasets | Split-Cifar100 | | 5-Datasets | |
|---|---|---|---|---|
| Metrics | $A_N(\uparrow)$ | $F_N(\downarrow)$ | $A_N(\uparrow)$ | $F_N(\downarrow)$ |
| UB | 90.85 | - | 93.93 | - |
| iCaRL | $81.38 \pm .65$ | $18.21 \pm .82$ | $83.35 \pm .29$ | $15.31 \pm 1.22$ |
| DER++ | $83.94 \pm .34$ | $14.55 \pm .73$ | $84.88 \pm .57$ | $10.46 \pm 1.02$ |
| Co2L | $82.49 \pm .89$ | $17.48 \pm 1.80$ | $86.05 \pm 1.03$ | $12.28 \pm 1.44$ |
| SI | $49.33 \pm .22$ | $35.45 \pm 1.45$ | $51.32 \pm .28$ | $32.35 \pm .16$ |
| EWC | $47.01 \pm .29$ | $33.28 \pm 1.17$ | $50.93 \pm .09$ | $34.94 \pm .07$ |
| LwF | $60.69 \pm .63$ | $27.77 \pm 2.17$ | $47.91 \pm .33$ | $38.01 \pm .28$ |
| L2P | $83.96 \pm .28$ | $6.32 \pm .38$ | $81.14 \pm .93$ | $4.64 \pm .52$ |
| DualPrompt | $86.51 \pm .33$ | $5.16 \pm .09$ | $88.08 \pm .36$ | $\mathbf{2.21 \pm .69}$ |
| CODA-Prompt | $86.25 \pm .74$ | $6.67 \pm .26$ | $83.24 \pm .59$ | $4.46 \pm .09$ |
| VPrompt(Ours) | $\mathbf{89.28 \pm .63}$ | $4.08 \pm .20$ | $\mathbf{90.74 \pm .28}$ | $3.97 \pm .30$ |

## 4.3 Evaluation Metrics

To assess the performance of continual learning methods, we employ two primary evaluation metrics: average accuracy and forgetting. Accuracy measures how well the model performs on all seen tasks after training up to the current point. Specifically, we report the average classification accuracy across all tasks at the end of training, which provides a direct indicator of the model's ability to retain knowledge while acquiring new information. In addition to accuracy, we use a forgetting measure to quantify the extent of performance degradation on previous tasks. Forgetting is computed as the difference between the maximum accuracy achieved on a task and the accuracy on that task after the final training phase. This metric is crucial for capturing catastrophic forgetting, one of the central challenges in continual learning, and is particularly important in scenarios where retaining previously acquired knowledge is as critical as learning new information. Together, these metrics offer a balanced evaluation of both plasticity (ability to learn new tasks) and stability (ability to retain old tasks), and are standard in the continual learning literature.

## 4.4 Implementation Details

Variational-based probabilistic prompts is a model-agnostic continual learning method that can be used for any transformer-based model. In this paper, we use the vision transformer model (ViT) Dosovitskiy et al. (2020) adopted by the previous lines of work in Continual Learning for vision recognition tasks. We use pretrained ViT model for the implementation of variational-based probabilistic prompts as the L2P Wang et al. (2022c) method, to compare with recent continual learning approaches.

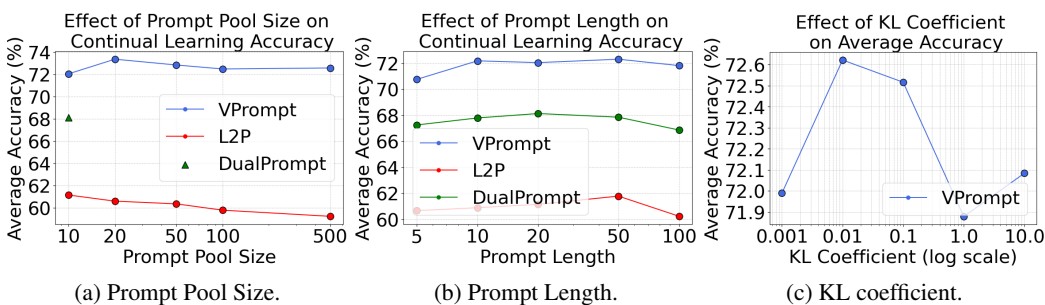

Figure 2: Ablation Analysis.

For ViT following Dosovitskiy et al. (2020), we use the representation of its first token $h_{[CLS]}$ as an image representation to predict the class of the input image $x$. Here [CLS] is a symbol that is encoded as a special beginning of a sentence, and $h$ is the whole input representation matrix from the ViT backbone encoder. We use a classification head as a linear transformation and a softmax function to obtain the classification probabilities over classes $c \in \{1, \ldots, \mathcal{C}\}$.

In the experiments we use cross entropy between the predicted classification probabilities and ground truth class labels to represent the classification loss. We use the Adam optimizer and set the batch size to 16. We set the learning rate to 0.03 and 0.005 varying for different datasets and experiments, $\beta_1 = 0.9$ and $\beta_2 = 0.999$. We set the prompt length from 5 to 20 tokens according to different datasets and experiments. In addition, we set the number of training epochs from 5 to 50 according to different datasets and experiments. For the code implementation, we use Pytorch Library Paszke et al. (2019) and HuggingFace Transformers library Wolf et al. (2019). All experiments are conducted on two Nvidia RTX A5000 graphic cards.

## 5 EXPERIMENTS

Table 2 compares performance of our variational inference-based probabilistic prompt method across 5, 10, 20 tasks with existing continual learning approaches for the ViT backbone model, including the previous SOTA: DualPrompt Wang et al. (2022b) and CODA-Prompt Smith et al. (2023). This benchmark on imagenet-r is attractive because the distribution of training data has significant distance to the pre-training data on imagenet, thus providing a fair and challenging problem setting. It is worth to notice that our method has strong gains in average accuracy across all three task lengths, with as much as 7%-8% improvement in average accuracy over CODA-Prompt. We also noticed that our method has a large performance gap between regularization-based approaches and even rehearsal-based approaches that used 500-5 000 replay samples. On the other hand, we can see from the table that our method suffers marginally slighter more forgetting than DualPrompt. We can regard this as the higher capacity that our method has for the learning. In fact, this is very reasonable and reflects the strength of our method. As we can see that with the increase of the tasks, the forgetting slightly increases but tends to be stable. In the whole scope, comparing with the regularization-based and the rehearsal-based approaches, our method has a much smaller forgetting rate.

Table 3 compares the performance of our probabilistic prompt method on Split-Cifar100 and 5-Datasets datasets with the existing continual learning approaches. These two benchmarks are attractive too because they include both diverse classes with various domain properties that have a large distance to the pre-training data on imagenet. We notice that our method has a strong performance that outperforms the SOTA-DualPrompt by average 3%. In the mean time, the forgetting rate that our method suffers is small comparing with the other approaches.

In Figure 2, we conduct several experiments to further analyze the performance of our algorithms against some hyper-parameter settings. Our aim is to conduct a detailed analysis on the capacity of our method to discover more insights. In Figure 2a, we plot the average accuracy while increasing the prompt pool size. We can see that for L2P the optimal pool size is 10: with the increase of the pool size the performance keeps dropping. For our method, the optimal pool size is 20, and it tends to

be saturated with similar performances. On the other hand, we can find that no matter how we adjust the prompt pool size, our method can consistently outperform the DualPrompt SOTA, showing that our method is highly competitive. In Figure 2b, we plot the average accuracy against the prompt length. We can see from the figure that by increasing the token length, the performance of our method also increases, and it also reaches a saturated point. Comparing with the other two baselines, our method has similar effect of prompt length on continual learning accuracy. In Figure 2c, we plot the effect of the KL coefficient on the average accuracy. We can see from the figure that the optimal KL coefficient is 0.01. This might be due to the trade-offs between the diversity of prompt selection and the task specific knowledge concentration: stronger regularization towards the KL term might make the posterior of latent distribution more diverse so that all the prompt components could be considered, which leads to more general knowledge sharing among the prompts but could not improve the performance of our method further. Weaker regularization of the KL term might lead to the latent distribution concentrating more on the specific prompt component. However the effect on the performance of our method is still limited.

## 6 CONCLUSION & DISCUSSION

In this paper, we introduce a novel framework called Variational Inference-Based Probabilistic Prompting (VPrompt) for rehearsal-free continual learning. Our approach reformulates prompt selection as a probabilistic inference problem, where the choice of prompts is modeled as a discrete latent variable conditioned on input features. Leveraging variational inference, our method estimates a posterior distribution over prompt selections and regularizes it with a uniform prior using KL divergence. This formulation enables uncertainty-aware prompt selection, allowing the model to flexibly adapt to new tasks while minimizing catastrophic forgetting, without requiring access to past data or task labels.

Our framework integrates seamlessly with frozen pre-trained Vision Transformers (ViT), requiring only a small pool of learnable prompts and achieving parameter-efficient continual learning. Through extensive experiments on Split CIFAR100, Split ImageNet-R, and a heterogeneous 5-dataset benchmark, our method demonstrates superior performance in terms of average accuracy and lower forgetting compared to existing state-of-the-art baselines, including CODA-Prompt, L2P, DualPrompt, EWC, and DER++. The probabilistic nature of our method particularly shines under domain shift and ambiguous task boundaries, where deterministic prompt selection strategies struggle.

Despite its strengths, our method has some limitations. The variational approximation assumes a fixed prior (uniform), which may not always capture the true dynamics of task relevance. Additionally, the method assumes a fixed-size prompt pool, which could become a bottleneck in extremely long task sequences or in cases requiring more granular prompt specialization. Furthermore, although we avoid using task labels, performance could be further enhanced with more structured priors or task-inferred guidance.

Looking forward, we plan to explore adaptive priors that evolve over time based on observed prompt usage, as well as hierarchical prompting to capture both coarse and fine-grained task features. Another promising direction involves integrating our framework with generative replay or self-supervised contrastive objectives to further improve performance in low-data or unlabeled continual learning settings. Finally, we aim to extend this approach to multi-modal continual learning scenarios and real-world edge deployments where memory and compute are highly constrained.

Our work takes a significant step toward making continual learning more scalable, flexible, and robust, especially in privacy-preserving or buffer-constrained environments.

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

## A    DETAILED DATASETS

**Split Cifar100**    The CIFAR100 dataset is a widely used image classification benchmark consisting of 60,000 color images (32×32 pixels), spanning 100 distinct object categories. Each class in CIFAR100 contains 600 samples, with 500 for training and 100 for testing.

In our experiments, we employ a class-incremental learning setting where the 100 classes are partitioned into 10 sequential tasks, each containing 10 mutually exclusive classes. The model is trained on one task at a time in a strict continual fashion without revisiting past task data. For compatibility with Vision Transformer (ViT) architectures, images are resized from 32×32 to 224×224. This benchmark provides a balanced, yet challenging setting for studying forgetting and knowledge transfer, particularly in the presence of subtle inter-class variations and dense object representations.

**Split Imagenet-R**    ImageNet-R is a curated subset of the standard ImageNet dataset, featuring 30,000 images from 200 ImageNet classes. What sets ImageNet-R apart is its use of non-naturalistic renditions of objects, such as sketches, cartoons, art, and paintings, to evaluate model robustness to distributional shifts and out-of-domain generalization. The dataset was introduced to assess how well models trained on natural images can recognize the same semantic categories rendered in atypical styles.

We split the 200 classes into 10 continual learning tasks of 20 classes each, preserving the class-incremental protocol used in CIFAR100. All images are resized to 224×224 to match ViT input size. This benchmark poses a significant challenge due to the visual mismatch between training and testing styles. Our probabilistic prompting mechanism, which incorporates uncertainty through variational inference, is particularly well-suited to handle such ambiguity in visual appearance across tasks.

**5-Datasets**    To further challenge the model's generalization capability, we evaluate it on a domain-incremental benchmark composed of five structurally diverse datasets: SVHN (Street View House Numbers): Derived from real-world house number images in Google Street View. It contains digit crops from natural scenes with significant background clutter, offering a noisy and complex digit recognition task. MNIST: A classic benchmark of handwritten digits in grayscale, MNIST is simple but highly structured. Each image is 28×28, representing digits from 0 to 9. CIFAR10: consists of 10 coarse-grained classes including animals and vehicles. It is used here to inject a natural image task amid digit-centric domains. NotMNIST: Composed of glyphs from typefaces representing letters A–J, NotMNIST is similar in format to MNIST but more visually diverse and slightly noisier. FashionMNIST: A drop-in replacement for MNIST, this dataset consists of grayscale images of clothing items, representing 10 categories such as shirts, shoes, and trousers.

## B    DETAILED BASELINES

**Learning to Prompt (L2P)**    Wang et al. (2022c): L2P introduces the idea of using a prompt pool in conjunction with a frozen Vision Transformer backbone, where a learned key-query mechanism selects task-relevant prompts. It achieves strong performance without accessing previous data, making it a compelling baseline for parameter-efficient continual learning. L2P serves as a primary baseline in our experiments due to its close methodological relationship with our approach.

**DualPrompt**    Wang et al. (2022b): An extension of prompt-based learning, DualPrompt decouples prompts into task-shared and task-specific components. The task-shared prompts capture common knowledge across tasks, while task-specific prompts specialize in preserving task identity. This

architecture provides greater flexibility in representing inter-task relationships and is particularly effective in settings with moderate task overlap.

**Elastic Weight Consolidation (EWC)**   Kirkpatrick et al. (2017): EWC is a classic method that regularizes updates to parameters that are deemed important for previous tasks. It approximates a Fisher Information Matrix to constrain future updates, reducing forgetting by preserving critical knowledge in the model weights.

**Learning without Forgetting (LwF)**   Li & Hoiem (2017): LwF mitigates forgetting by enforcing output consistency with the model's previous predictions. This method uses knowledge distillation loss from the model trained on earlier tasks to maintain performance, making it applicable even in the absence of task boundaries or access to old data.

**Synaptic Intelligence (SI)**   Zenke et al. (2017): Similar to EWC, SI accumulates importance scores for each parameter during training and applies path-integral-based regularization to slow down updates to those weights. It is particularly efficient and well-suited for online settings.

**iCaRL (Incremental Classifier and Representation Learning)**   Rebuffi et al. (2017): A rehearsal-based method that stores exemplars from past tasks and combines nearest-class-mean classification with incremental feature learning. Though it uses memory, it provides a strong benchmark for hybrid strategies.

**DER++ (Dark Experience Replay)**   Buzzega et al. (2020): DER++ stores logits of previous tasks along with exemplars, enabling knowledge distillation and replay-based learning. While not rehearsal-free, it performs very competitively in class-incremental learning.

**Co2L (Contrastive Continual Learning)**   Cha et al. (2021): A recent contrastive learning-based approach that uses self-supervised losses to preserve representations across tasks. Co2L is effective in both supervised and unsupervised continual learning.

## C   DETAILED RELATED WORK

### C.1   PROMPT TUNING AND ADAPTER METHODS

Recent advances in transfer learning have highlighted the efficiency of *prompt tuning*, which has emerged as a powerful alternative to full model fine-tuning. Originally developed for NLP, prompt tuning techniques such as prefix-tuning Li & Liang (2021) and soft prompt tuning Lester et al. (2021) train small, continuous prompt vectors while freezing the rest of the model. This paradigm has been successfully extended to vision transformers (ViTs), offering a parameter-efficient mechanism for domain adaptation in computer vision.

Closely related are *adapter methods* Houlsby et al. (2019), which introduce small trainable bottleneck modules between transformer layers. Adapters enable selective tuning without modifying the backbone model, thus reducing computational cost and mitigating overfitting. Compared to adapters, prompt tuning is often more lightweight and modular, lending itself naturally to continual learning settings where rapid, task-specific adaptation is critical.

### C.2   UNCERTAINTY IN CONTINUAL LEARNING

Incorporating uncertainty is essential for robustness in CL, especially under ambiguous task boundaries and limited data. Bayesian Neural Networks model parameter uncertainty using posterior distributions, while Variational Continual Learning (VCL) Nguyen et al. (2017) uses variational inference to approximate posterior beliefs over weights. Ensemble methods and dropout approximations have also been employed to estimate prediction uncertainty in CL.

Existing uncertainty-based methods focus primarily on the model parameters or output predictions. In contrast, our work introduces uncertainty at the prompt selection level. By using variational inference to learn a posterior distribution over prompts, we provide a lightweight and scalable mechanism

for uncertainty-aware task conditioning. This approach enhances the model's ability to generalize and reason under ambiguity without necessitating full Bayesian modeling of the entire network.

### C.3 PROBABILISTIC PROMPTING

Our method introduces a novel probabilistic prompting framework for continual learning. We formulate prompt selection as a latent variable problem and use variational inference to approximate the posterior distribution over prompt selections. The learning objective includes a KL-divergence regularizer that penalizes deviation from a prior distribution, encouraging diverse exploration of the prompt space during early training and promoting consolidation as learning progresses.

This probabilistic perspective enables the model to manage uncertainty in prompt-task alignments, leading to improved robustness and reduced overfitting. Unlike L2P's deterministic mechanism, our approach allows the model to hedge between multiple prompt candidates, which is especially beneficial in task-agnostic or non-i.i.d. settings. Empirical results confirm that this method enhances stability and flexibility, making it a compelling alternative for scalable, uncertainty-aware continual learning.

