# OpenReview forum: "Variational Inference based Probabilistic Prompt for Rehearsal-Free Continual Learning"
_ICLR.cc/2026/Conference — ICLR 2026 Conference Withdrawn Submission_

### Official Review · Reviewer_adB5 · 2025-10-19

**Soundness:** 2
**Presentation:** 3
**Contribution:** 2
**Rating:** 2
**Confidence:** 4

**Summary:**

This paper introduces VPrompt, a variational inference-based probabilistic prompting framework for rehearsal-free continual learning (CL). Unlike deterministic prompt selection methods such as L2P or DualPrompt, VPrompt models prompt selection as a latent variable inference problem. Specifically, it learns a posterior distribution $q(z|x)$ over prompts conditioned on inputs, regularized by a uniform prior via a KL-divergence term. This probabilistic design enables uncertainty-aware prompt selection, mitigating overconfidence and improving generalization under ambiguous or non-stationary task transitions. Experiments across Split CIFAR100, Split ImageNet-R, and a five-dataset benchmark show consistent performance gains over both deterministic prompt-based and rehearsal-based baselines (e.g., CODA-Prompt, DualPrompt, DER++).

**Strengths:**

- **Conceptual Novelty and Clarity:** The paper contributes a clear probabilistic formulation for prompt selection, addressing a genuine limitation in deterministic prompt-tuning methods. By grounding the design in variational inference, the method brings a rigorous probabilistic perspective to prompt-based continual learning, connecting CL literature with recent advances in variational prompt tuning (cf. VPT, ICLR 2023).
- **Strong Empirical Results:** Consistent improvement over competitive baselines (L2P, DualPrompt, CODA-Prompt) on multiple benchmarks is convincing. The improvement on ImageNet-R is especially notable, supporting claims of robustness under domain shift.
- **Rehearsal-free, parameter-efficient design** The proposed approach remains buffer-free and does not require task labels or additional storage, which is important for real-world CL use cases where privacy or storage is constrained. Integration with frozen ViTs ensures fair comparison with prior prompt-based methods.
- **On generalization.** Modeling the posterior over prompts allows the model to represent task ambiguity, aligning well with Bayesian continual learning principles. The framework bridges ideas from VPT (ICLR 2023) and probabilistic reasoning, offering interpretability in prompt usage patterns.
- **On ablations** Hyperparameter sensitivity and qualitative discussion (e.g., on the KL coefficient) demonstrate thoughtful empirical investigation.

**References**

[1] Derakhshani *et al.* "Variational prompt tuning improves generalization of vision-language foundation models." Me-FoMo workshop, ICLR 2023.

**Weaknesses:**

- **Theoretical understanding:** While the paper borrows the ELBO formulation, the theoretical motivation for how this probabilistic modeling specifically mitigates forgetting could be deepened. Connections to Variational Continual Learning (Nguyen et al., 2017) and Bayesian CL literature are mentioned but not rigorously analyzed.

- **Lack of comparisons to relevant literature:** What concerns me is the missing comparisons and discussions with directly related works such as CLAP4CLIP [1], which is a probabilistic continual learning method designed with prompt-based support. For instance, the authors could have compared or discussed how their variational latent modeling differs from contrastive latent regularization in CLAP4CLIP.

- **Limited Evaluation Scope:** All experiments are conducted with ViT backbones; no results are shown for multi-modal or language-vision settings, where probabilistic prompting might show even greater benefits. No comparisons are made to variational prompt tuning in generalization (e.g., VPT), which is the most direct precursor method in terms of probabilistic prompt learning.

- **Missing time complexity analyses:** Runtime or computational cost comparisons (e.g., training time vs. L2P/DualPrompt) are absent, which would be valuable for assessing scalability.

- **Ablation depth** The ablations are somewhat limited to scalar hyperparameters; qualitative analyses (e.g., prompt selection entropy across tasks, prompt re-use visualization) would strengthen the argument about uncertainty modeling.

**References:**

[1] Jha *et al.* "CLAP4CLIP: Continual Learning with Probabilistic Finetuning for Vision-Language Models." NeurIPS 2024.

**Questions:**

- Have the authors explored updating the prior $p(z)$ over prompts as a function of past prompt usage or task recency? A fixed uniform prior might underutilize information about task correlations.
-  Since the prompt pool size is fixed, how does the model behave under 50+ tasks? Is there evidence of saturation or collapse in the posterior distributions?
- Can the authors provide examples or visualizations where uncertainty-aware selection $q(z∣x)$ leads to different predictions than deterministic selection?
- Given recent successes in multimodal prompt tuning, could VPrompt be integrated with CLIP or BLIP backbones, and how might uncertainty at the prompt level interact with cross-modal alignment?

---

### Official Review · Reviewer_vyYV · 2025-10-29

**Soundness:** 3
**Presentation:** 3
**Contribution:** 3
**Rating:** 4
**Confidence:** 1

**Summary:**

This paper tackles the problem of rehearsal-free continual learning, where models must learn new tasks sequentially without retaining samples from previous tasks. The authors introduce VPrompt, a probabilistic prompt learning method that replaces deterministic prompt selection with a variational inference framework. Instead of hard-assigning prompts to tasks, the model learns a distribution over prompts conditioned on the input, balancing a classification term and a Kullback–Leibler regularization term. This allows the system to capture uncertainty during task transitions and reduce catastrophic forgetting. The results on several standard continual learning benchmarks show steady improvements over prior prompt-based approaches. The contribution is technically solid and well-motivated, though more incremental than transformative.

**Strengths:**

- The motivation of introducing probabilistic modeling into prompt selection is clear and relevant for continual learning.
- The variational inference formulation is principled and integrates smoothly with prompt-based architectures like L2P and DualPrompt.
- The method remains lightweight and rehearsal-free by freezing the backbone and updating only prompts, keys, and the classifier head.

**Weaknesses:**

- The method's novelty appears somewhat limited. Using variational inference for uncertainty modeling or soft selection is a common technique, and its application to prompt selection in CL, while new, feels like an incremental methodological contribution.

- The paper fails to compare against several more recent and relevant methods. For example, comparisons with C-Prompt or APT seem to be missing, which makes it harder to situate the paper's contribution relative to the current state of the art.

- The paper lacks analytical depth to fully explain why the method works. There is no visualization or statistical analysis of the learned posterior $q(z|x)$. This makes it difficult to verify the 'uncertainty awareness' claim, as we don't see if ambiguous inputs truly lead to higher-entropy distributions.

**Questions:**

1. Given that variational inference is a well-established technique, could the authors further elaborate on the specific methodological novelty of VPrompt beyond applying VI to the problem of prompt selection?

2. The paper is missing comparisons to other recent prompt-based CL methods like C-Prompt and APT. Could the authors discuss how VPrompt is positioned relative to these methods and provide a rationale for their exclusion from the experiments?

3. Could you provide a deeper analysis or visualization of the learned posterior distribution $q(z|x)$? For instance, how does the entropy of this distribution change for inputs that are ambiguous or lie at task boundaries? This would be crucial to validate the 'uncertainty awareness' claim.

---

### Official Review · Reviewer_T3nj · 2025-10-31

**Soundness:** 2
**Presentation:** 2
**Contribution:** 2
**Rating:** 4
**Confidence:** 5

**Summary:**

This paper proposes VPrompt, a rehearsal-free continual learning method that frames prompt selection as a probabilistic inference problem using variational inference. The authors claim that modeling uncertainty over prompt assignments improves robustness to domain shift and task ambiguity, while eliminating the need for replay buffers or task identifiers. Empirical results on Split CIFAR100, Split ImageNet-R, and a 5-dataset sequence show consistent improvements over existing prompt-based and rehearsal-free baselines.

**Strengths:**

1. The integration of uncertainty modeling into prompt selection is conceptually appealing and aligns with recent interest in probabilistic continual learning.
2. The method is parameter-efficient and compatible with frozen Vision Transformers, which is practical for deployment-constrained settings.
3. Experimental evaluation covers diverse and challenging benchmarks, including significant domain shifts.

**Weaknesses:**

1. The proposed “variational inference” framework reduces to a standard softmax-based soft routing mechanism with a KL-divergence regularizer toward a uniform prior. This bears little resemblance to canonical variational inference (e.g., in VAEs or Bayesian neural networks), where latent variables are stochastic and inference involves sampling or structured posteriors. Here, the posterior $q(z∣x)$ is deterministic given $x$ , and no sampling or marginalization over latent prompts is performed during inference or training. Consequently, the use of variational inference appears largely terminological rather than methodological.
2. The paper omits comparison with genuine Bayesian or uncertainty-aware continual learning approaches, such as Variational Continual Learning, deep ensembles, or Monte Carlo dropout–based methods. Without such comparisons, the claimed advantage of  uncertainty-awareness remains unsubstantiated. Moreover, the reported superiority over rehearsal-based methods like iCaRL or DER++ is misleading, as those methods operate under a more constrained memory budget (e.g., 20 exemplars per class), whereas VPrompt uses a fixed prompt pool whose memory cost is not fairly normalized against replay buffers.
3. The ablation studies (Figure 2) only report average accuracy without examining how the KL coefficient or prompt pool size affects forgetting, prompt entropy, or inter-task interference. Crucially, the paper provides no qualitative evidence, such as prompt usage distributions across tasks or entropy heatmaps, to demonstrate that the model indeed leverages uncertainty in ambiguous regions. The claim that soft prompt routing smooths decision boundaries remains speculative without such analysis.
4.  All experiments assume task-incremental learning with known task boundaries at test time (i.e., the model knows which label subset to predict). However, the abstract and introduction emphasize applicability in task-agnostic or ambiguous” scenarios. The method’s performance under true class-incremental or task-agnostic settings—where the model must infer both task and class—is not evaluated, casting doubt on its real-world utility.
5. While VPrompt avoids storing raw data, the prompt pool itself functions as a form of implicit memory that grows with task complexity (e.g., 20 prompts × 20 tokens × 768 dimensions ≈ 614K parameters). This is not meaningfully “buffer-free” from a memory-efficiency standpoint, especially when compared to methods like EWC or SI that require no additional storage beyond model weights.

**Questions:**

pls see above

---

### Official Review · Reviewer_4vKy · 2025-11-01

**Soundness:** 2
**Presentation:** 3
**Contribution:** 2
**Rating:** 2
**Confidence:** 4

**Summary:**

To address the reliance on deterministic selection lacking uncertainty modeling in existing prompt-based Continual Learning (CL) methods，this paper introduces a Variational Inference based Probabilistic Prompt (VPrompt) framework. VPrompt models prompt selection as a stochastic latent variable, enabling soft prompt routing and actively promoting the reuse of prompts in the prompt pool. Extensive experiments are conducted on Split CIFAR-100, Split ImageNet-R, and a 5-dataset benchmark.

**Strengths:**

1.The manuscript is well-written and easy to understand, helping readers grasp the proposed method and results effectively.

2.Figures and tables are clearly presented and well-organized

**Weaknesses:**

1.The proposed method does not effectively achieve the stated motivation, such as introducing uncertainty and addressing the limitation of the model’s capacity to handle ambiguous or overlapping tasks.

2.The overall innovativeness of the framework is relatively limited because the method is essentially a standard application of the variational Inference paradigm to prompt learning.

3.The method proves to be ineffective in alleviating catastrophic forgetting, based on the comparative results in Tables 2 and 3. Its performance on average forgetting is shown to be inferior to prior methods.

**Questions:**

1.The authors ought to clarify how the proposed method “explicitly modeled via the entropy of $q(z|x)$”. From the formulation, the entropy term seems implicitly included in the KL divergence but is not separately modeled or optimized in the objective.

2.Since the KL term encourages $q(z|x)$ to approach a uniform distribution, the authors should explain whether this would reduce the model’s discriminative ability or oversmooth the prompt distribution.

3.The authors should discuss the efficiency of their method compared with other prompt-based approaches. The variational expectation necessitates M (the size of prompt pool) full forward passes for every input. This could introduce substantial computational overhead.

4.The authors must explain how variational probabilistic prompting protects knowledge; currently, the method only appears to introduce stochasticity. Its weakness is empirically confirmed by the higher average forgetting $F_N$, observed on Split-ImageNet-R and 5-Datasets compared to several baselines, indicating weaker resistance to forgetting.

---

### Note · Authors · 2025-11-19

I have read and agree with the venue's withdrawal policy on behalf of myself and my co-authors.